# Exploring Parents’ Concerns Regarding Long-Term Support and Living Arrangements for Their Children with Fragile X Syndrome

**DOI:** 10.3390/genes13091654

**Published:** 2022-09-15

**Authors:** Kaylynn Shuleski, Laura Zalles, Reymundo Lozano

**Affiliations:** 1Department of Genetics and Genomic Sciences, Icahn School of Medicine at Mount Sinai, New York, NY 10029, USA; 2Department of Pediatrics, Icahn School of Medicine at Mount Sinai, New York, NY 10029, USA

**Keywords:** Fragile X Syndrome, FXS, intellectual disability, autism spectrum disorder, living arrangements, caregiver

## Abstract

Given limited data regarding future planning specific to Fragile X Syndrome (FXS) individuals and the growing population of individuals within this community, this study sought to explore the concerns and challenges caregivers of individuals affected by FXS encounter when considering long-term support plans. This involved identifying the reasons individuals with FXS continue to reside with family and the reservations caregivers have regarding future supports and living arrangements. We administered an anonymous online survey consisting of 34 questions assessing eligibility, living arrangements/supports, and future concerns. We found that most individuals with FXS were affected with moderate Intellectual and Developmental Disabilities (IDD) and co-occurring behavioral conditions but had overall good health. The majority of individuals with FXS currently resided with family due to parental desire, their own desire, and the inability to live independently. For one-third of caregivers, the plan for future living arrangements is to continue residing with family members long-term. A large proportion of caregivers had not considered alternative arrangements or were unsure. More than 70% of caregivers of individuals with FXS are concerned about multiple aspects of the individual’s future. Caregivers of younger individuals are the most concerned, but also believe they have time before they need to plan or are unable to currently assess the future need for support.

## 1. Introduction

Fragile X Syndrome (FXS) is the leading inherited cause of intellectual disability (ID), with affected individuals often displaying a range of mild to severe ID as well as developmental delay [1]. The disorder is caused by the expansion of the CGG trinucleotide repeat of the fragile X mental retardation 1 (*FMR1*) gene, located on the X chromosome [2]. The normal *FMR1* allele typically has 5 to 44 CGG repeats, whereas individuals with FXS have full mutation alleles of >200 CGG repeats. Expansion of the repeat leads to silencing of the gene and subsequent loss of protein production, which plays an important role in the development and maintenance of neuronal synaptic connections [3]. Loss of this protein results in an imbalance of inhibitory and excitatory neuronal circuits which is thought to be the underlying cause of the neurologic clinical manifestations of FXS. 

Given the location of the *FMR1* gene, FXS follows an X-linked inheritance pattern which corresponds to the observed incidence of the disorder: approximately 1 in 4000 males and 1 in 7000 females [4]. About 90% of males with a full FXS mutation display ID. Females with FXS have lower rates of ID due to X inactivation and environmental influence, however, still show cognitive impairment when compared to their age matched peers. [5]. FXS also has a high comorbidity rate of autism spectrum disorder, with an incidence of 50–80% [6]. FXS is recognized as the most common monogenetic cause of ASD, with affected individuals displaying cognitive and behavioral impairments [6]. 

Common medical conditions that present in adolescents with FXS include otitis media, seizures, sleep disturbance, and gastrointestinal disorders [7]. These issues are generally non-life-threatening and seem to resolve with age, with affected individuals having normal lifespans. Most individuals with FXS will require lifelong care or support due to the intellectual and developmental delays associated with their disorder. The relatively high incidence rate of FXS, along with improved clinical diagnostics and average lifespans, makes this a growing population of individuals clinically diagnosed with ID.

A national survey found that of adults with FXS, 51% of females and 70% of males still co-reside with parents [8]. Another study found that most people with ID wish to “age in place”, meaning they want to continue living in their family home when they reach old age [9]. This poses difficulties, because as adults with FXS continue to age, their caregivers, who are often their parents, may be reaching elderly age. As these caregivers age, their health may naturally decline, making it difficult to provide the constant care and support needed by their adult children. As a result, parents have expressed serious concerns about long-term care and possible lifelong living arrangements for their children with ID [9,10]. 

A prior study found a positive relationship between the functional level of adults with FXS and independent residential settings [8]. An individual’s level of functional skills was the strongest predictor of overall independence. Another study found that the presence of co-occurring mental health concerns such as hyperactivity, aggressiveness, anxiety, depression and/or ASD is associated with less independence in adult life [11]. These individuals may require placement in less independent residential settings, such as group homes, yet are often rejected due to the added challenges associated with their co-occurring mental health issues. Lack of functional skills and the presence of comorbid mental health concerns act as barriers to independent living, making these important areas to address in the care of individuals with FXS. 

Additionally, many families that care for children with ID face financial burdens due to disorder-associated medical costs or loss of potential income. One study found that 52% of caregivers of children with ASD/ID reported financial difficulty, with 51% having to stop work to care for their child [12]. Although there are some resources available to families of children with disabilities within the United States, such as Medicaid, waiver services, or supplemental security income, they only cover a fraction of the costs associated with long-term caregiving. In 2011, the average annual per person cost for state operated Intellectual and Developmental Disability (IDD) residential services was $274,173 [13]. The costs of these expensive residential services may make it difficult for parents to justify placing their children in these facilities when they can continue to care for them in the family home. Although studies have shown that long-term caregiving can be detrimental to the caregiver’s health, parents may forfeit their own health to avoid costly out-of-home services for their children [14]. 

As parents struggle to care for their children with FXS, they will ultimately seek additional assistance or alternative residential placements for them. Parents have expressed both interest in being involved in their child’s lifelong care and apprehensions about the quality of care available. Because of this, caregivers may want to start planning early and gaining information about all possible options and supports. Unfortunately, most published data pertains to ASD or ID, while information regarding FXS families is limited. Although ID is often diagnosed in individuals with FXS, and ASD is a highly comorbid disease, these are separate disorders that have clinical features unique to each of them. Different medical concerns or barriers to independent living may need to be considered when planning future long-term care for people with FXS. It is important to identify these challenges and understand how families currently address them, as their decisions could serve as a precedent for others. 

The goal of this study is to identify the concerns and challenges caregivers of individual with FXS encounter when considering long-term support plans for their affected loved ones. This includes identifying the reasons individuals with FXS reside at home, the specific concerns caregivers have regarding future supports/living arrangements, and the barriers they face when planning for the future. Information regarding financial aid, long-term supports, and perceptions surrounding out-of-home placements will be useful for future families affected by FXS, policy makers, and health practitioners.

## 2. Materials and Methods

The newly generated, anonymous survey (Appendix A) was administered online through REDCap survey software. Participants had to be caregivers of individuals with FXS, caused by a full mutation of the *FMR1* gene, to be included in the study. The project was approved by the Icahn School of Medicine at Mount Sinai Institutional Review Board (IRB-20-03817). Survey data was collected from 26 October 2020 to 28 February 2021. 

The survey was sent to members of the National Fragile X Foundation (NFXF), which supports families affected by FXS throughout North America. The survey was also sent to a subset of families registered with NFXF Research Listserv; those interested in research opportunities received an email invitation to participate in the study. A short description of the study and a link to the survey was also hosted on the NFXF MyFXResearch Portal.

The survey included a total of 34 questions and was divided into six sections: eligibility, survey respondent demographics, individual with FXS characteristics, living arrangements/supports, future concerns, and a free response portion. The eligibility portion ensured survey respondents were caregivers of individuals with a full mutation of the *FMR1* gene. The FXS individual characteristics portion included the Waisman Activities of Daily Living (W-ADL) scale, a 17-item scale designed to assess daily living skills in individuals with ID [15]. The free response portion provided participants the opportunity to describe concerns and challenges that were not included in the survey, as well as their experiences addressing these concerns. 

The survey was validated by the NFXF Research Readiness Program, a group of experts in the field of FXS to ensure the questions were appropriate and accurate from both a research and clinical standpoint. Following a revision process, the survey was piloted through the NFXF volunteer group, which invited current caregivers to complete the survey and provide feedback. This was to ensure the survey accurately addressed aspects important to the community in a manner that was generally comprehensible. 

Microsoft Excel version 16.47.1 was used to perform descriptive statistics of categorical variables which are presented as frequencies. SAS University Edition version 3.8 statistical software was used to perform statistical analysis of variables and in the determination of non-parametric data. Survey responses from questions B6 and D1, presented as Likert-scales, were converted to numerical values and the mean for each response was calculated to provide a *Concern Average* score and a *Dependence Average* score, respectively. This calculation converted the data into a continuous variable and allowed for further statistical analysis. Fisher’s exact test was used for categorical variables and small sample sizes. One-way non-parametric ANOVA was used for categorical and continuous data. Spearman rank correlation coefficient was used for comparison of non-parametric ordinal data. All statistical tests were two-sided and a *p*-value of <0.05 was used in the determination of statistical significance. A false discovery rate was not calculated given the cross-sectional study design. Responses to free response questions were analyzed qualitatively to identify common themes.

## 3. Results

A total of 81 surveys were initiated. Survey responses that were incomplete and did not meet inclusion criteria were excluded from the study, leaving 63 eligible responses for analysis. 

### 3.1. Survey Respondents

Of the 63 respondents, caregivers were predominantly mothers, Caucasian, and provided care to one individual with FXS (Table 1). The regional spread of respondents was equally distributed across North America, with two responses from outside of the U.S., both of which were from Canada. The majority of respondents were college educated, with 36.5% of whom completed a Bachelor’s degree and 30.2% of whom completed a post-graduate degree. Annual household income was representative across various socioeconomic statuses. 

### 3.2. Individuals with FXS Characteristics

Of the 63 survey responses which provided details about the individuals with FXS, individuals were predominantly male, between the ages of 15–34 years, and affected with moderate IDD. Only one individual was over the age of 45 years. Most individuals were affected with at least one co-occurring mental health condition, with anxiety, attention disorders, and hyperarousal being the most frequently reported. Out of 62 individuals, 61.3% were affected with ASD, which is consistent with previously reported incidence [6] The proportion of individuals with moderate or severe IDD, hyperarousal, hyperactivity, and ASD were higher in male individuals with FXS than female individuals with FXS. The overall health of the study population was reportedly good or excellent, with only one caregiver reporting fair and none reporting poor or very poor for overall health. 

### 3.3. Living Arrangements

Most individuals with FXS (84%) currently resided at home with parents or other family members, while the remaining 16% of the study population resided in alternative, out-of-home living arrangements (Appendix B). 

When asked how much certain factors apply to the reason the individual with FXS resides at home, parental desire (72.5%), individual desire (65.4%), and the inability to live independently (71.2%) were a major factor for many (Figure 1). Less frequently reported reasons were dissatisfaction with available services, not having considered other options, and limited residential options. Rejection from a residential placement was only reported once (1.9%) as the reason for why an individual resided at home. 

When asked what future living arrangement the caregiver has considered most for the individual with FXS, 33% selected, “continue living at home with family members long-term.” When compared to current residencies, a greater percentage of caregivers were considering out-of-home placements for future living arrangements. There were 19% of caregivers who selected “Not yet considered/Unsure.”

Figure 2 displays seven different reasons caregivers have not started planning or considered other living arrangements. For most reasons, the majority of respondents did not agree the reason was applicable to them. The most frequently (27%) reported reason was “Not a concern yet/still have time,” while the second most was “Other” (17%). Specific themes cited as “Other” included distrust with residential services and the reluctance to consider an arrangement that would prevent the individual from residing with family members.

There were 17 caregivers who selected “Not a concern yet/still have time” as a reason they had not started planning or considered other living arrangements (Figure 2). When assessing the age groups of these individuals, the most frequent (47%) was caregivers of individuals < 15 years of age. The proportions would also decrease with increasing age group, meaning the caregivers who selected “Not a concern yet/still have time” were more likely to care for a younger individual with FXS. 

### 3.4. Future Concerns

Caregivers were provided a Likert-scale to rank their level of concern regarding six aspects of the individual’s future: (1) Who will care for my child, (2) Financial support, (3) Availability of resources, (4) Quality of resources, (5) Transition to supports/living arrangements, and (6) Abuse (Figure 3). Overall, caregivers were moderately/extremely concerned regarding all six aspects of the individual’s future. Abuse had the lowest overall concern while the quality of resources had the highest overall concern. 

The level of concern regarding future financial support was separated by three different financial factors and analyzed using Fisher’s exact test. The factors were (1) annual household income, (2) financial resources (i.e., public funding, special needs trust, savings account, etc.), and (3) a designated financial guardian. Regardless of the annual household income, the number of financial resources the caregiver had in place, or if there was a predetermined financial guardian, caregiver’s level of concern regarding future financial support remained moderate/extreme.

The level of concern for individual aspects were separated by age group and analyzed using Fisher’s exact test. Overall, caregivers remain concerned about the future, but the level of concern for certain aspects were observed to significantly decrease with increasing age of the individual with FXS. Caregivers of individuals with FXS who were less than 15 years old were shown to have a statistically significant increased concern compared to those of individuals with FXS greater than 35 years when asked about (1) Who will care for my child (*p* = 0.003), (2) Financial support (*p* = 0.015), (3) Availability of resources (*p* = 0.032), (4) Quality of resources (*p* = 0.014), and (5) Abuse (*p* = 0.001). For one aspect, Transition to support/living arrangements, the level of concern did not significantly decrease with age (*p* = 0.065).

The concern average score was compared to the dependence average score and analyzed using Spearman rank correlation. A small, but statistically significant positive correlation (R^2^ = 0.1263, *p* = 0.0043) was observed between increasing dependence and increasing concern, with the overall dependence average accounting for approximately 12% of the variation observed in the concern average. 

The concern average score was compared to both the level of reported IDD and the number of co-occurring mental health conditions in the individual with FXS. Analysis using one-way non-parametric ANOVA showed that regardless of the severity of IDD or the number of co-occurring mental health conditions, there was no major difference in the level of caregiver concern.

A smaller subset of respondents, those considering out-of-home living arrangements for the future, were analyzed further. Fisher’s exact test was used to compare the concern average score between those who have started planning and those who have not. This showed that regardless of whether the caregiver started planning, the level of concern amongst the two groups was not different (*p* = 0.22).

### 3.5. Free Response

Among the free responses asking for additional concerns regarding long-term supports/living arrangements, a few recurring themes included a general fear of the unknown, concerns surrounding the individual with FXS outliving the appointed long-term caregiver, and the limitation that no amount of financial support can guarantee security with the future. These concerns were expressed in the following specific quotes: 


*“For us there are just a lot of unknowns… good chance he will become self sufficient, however we are prepared for him to live at home for the remainder of his life as well.”*



*“Fear of the unknown and her ability to adapt to changes without supervision.”*



*“We don’t know what his abilities will be when he is an adult, so we cannot fully plan at this stage.”*



*“What happens if he outlives the caregivers we choose for him (other siblings)?” B) “Person with FXS outlives successor caregiver.”*



*“Never enough money to feel comfortable”*


While there were no clear, recurring themes identified among the experiences shared expressing these concerns, the variety of the responses strongly represent the breadth of struggles caregivers experience when planning. Specific quotes as follows: 


*“We are not ready to let him go”*



*“Like nailing jello to a wall.”*



*“A nightmare”*



*“I honestly have not addressed the what if scenario for when me and his father are no longer around… no siblings and this is something that I often think about but am unsure of when or what I even need to have arranged due to the unknowns.”*


## 4. Discussion

Given the limited data regarding future planning specific to individuals with FXS and the growing population of this community, this study sought to explore the concerns and challenges caregivers of FXS individuals encounter when considering long-term support plans. This involved identifying reasons individuals affected by FXS continue to reside with family and the reservations caregivers have regarding future supports and living arrangements. Most of the individuals within the study population currently resided with parents or family members due to parental desire, their own desire, or the inability to live independently. While the inability to live independently could be due to an individual’s lack of functional ability, it is also possible that these are younger individuals whom regardless of their functional ability would not be living independently at their age. Other reasons, such as dissatisfaction with available resources, limited residential options, and the caregiver not considering other options, were less often the reason the individual with FXS resided with family. Given that these factors were cited less often shows that a large sample of respondents have not considered alternative living arrangements, thus making it difficult to assess the challenges caregivers face when planning and how they overcome said barriers. 

For future living arrangements, the most considered option was for the individual to continue residing with family members long-term. The frequency of this choice, and the lack of consideration for out-of-home living arrangement highlighted by the previous responses, provides insight to the consensus that caregivers hope the individual with FXS will remain under the care of family members long-term. While caregivers are considering out-of-home living arrangements for the future, a large proportion had not considered alternative arrangements or were just unsure. A major reason why these caregivers had not considered different arrangements was because they did not think it was a concern yet or believed they still had time, while other factors such as dissatisfaction with the available options or the reluctance to consider any arrangement without a family member were also noted. Interestingly, when exploring the responses of caregivers who believed they had time before considering options, we found that they were more likely to care for a younger individual with FXS. This is consistent with the idea that caregivers of young individuals are more likely to be younger adults themselves. This means these caregivers have more perceived life years to provide support and more time to consider future plans, hence why they have not given it as much thought compared to caregivers of older individuals with FXS. 

We found that caregivers are generally concerned about multiple aspects of the individual affected by FXS’s future. Factors such as the severity of IDD and the number of co-occurring behavioral conditions did not have an impact on the overall level of concern experienced by caregivers. However, when we explored individual concern aspects and compared them to age groups, we saw the level of caregiver concern for five out of six factors decreased as the age group of the individual affected by FXS increased. The only factor where the level of concern did not decrease was regarding the individual with FXS’ transition to different supports and living arrangements. This result stresses the sentiment held by caregivers, that regardless of the amount of planning they do or the number of supports in place, they are unable to predict how the FXS individual will adjust to these transitions. This concern highlights the importance of establishing resources early in the individual’s life as it provides time for both the individual with FXS to adjust as well as the caregiver to make any necessary accommodations. 

There was also an observed decrease in the caregiver’s concern regarding the future as the individual with FXS’ overall dependence decreased. This shows that caregivers of more independent individuals, or those who have achieved set functional skills, are less likely to be concerned about the future. This underscores the importance of early intervention and establishing support for individuals affected by FXS, so they can work to attain skills early in life with hopes of reaching greater independence. Another interesting result was that the level of concern regarding future financial support did not differ based on the amount of annual household income or the number of financial resources put in place. This finding is well represented by the free response quote, “Never enough money to feel comfortable”, showing that no amount of financial support can fully eliminate a caregiver’s concern. This may represent the idea that it is not a question about the cost of currently available resources, but rather the quality, quantity, and availability that needs to be addressed. 

To summarize, while caregivers of younger individuals are generally more concerned about the future, they are less likely to have considered future supports and living arrangements. While it is possible that early planning may help to alleviate some concern, the free response quotes highlight the idea that it is a general fear of the unknown or inability to predict how the FXS individual will develop that leads to anxiety. It is possible that for these younger individuals, caregivers are still working to understand their FXS diagnosis and how it will impact them as adults, resulting in the inability to plan for the future. While caregiver concern may decrease as the FXS individual ages, establishing early interventions and helping the individual attain functional skills may help to reduce some concerns regarding the future. 

A major goal of this study was to understand the challenges caregivers face when planning alternative supports and living arrangements. Given the small study population of individuals who did not reside with family, we were limited in our ability to assess this measure. While the W-ADL scale is a well-validated measure in the field of research for ID, its calculation and utilization as a dependence average score for this study has never been done before, possibly limiting the validity of the statistical analyses where the score was used. The different characteristics of the individuals with FXS, such as IDD severity, presence of co-occurring conditions, and overall health score were all caregiver reported. This means the reported characteristics may be based on caregiver perceptions rather than true clinical diagnoses, which could limit clinical validity and generalizability of the study.

Our analysis was limited by the cross-sectional nature of a survey study. This was a questionnaire-based survey for Fragile X Syndrome care providers. We relied on the self-reporting of survey patients and therefore were unable to confirm molecular diagnosis for the individuals with FXS. Our sample size was limited and included both male and female individuals with FXS.

One future direction for the study could be to explore the community of individuals currently residing in out-of-home living arrangements. It would be helpful to explore the factors considered and the challenges faced by these caregivers when they were searching. It would also be interesting to assess these caregivers’ concerns given that the individual is in an established residential setting. Future studies could also distribute the survey to both caregivers of non-FXS related individuals with IDD and individuals affected by FXS. This could identify specific differences between these two populations, as they are often equated in research without the consideration of common FXS-related comorbidities. We would also hope to survey a larger and more ethnically diverse population as cultural factors and social determinants of health likely play an important role in the FXS community.

## 5. Conclusions

Of the study population, we found that most individuals with FXS were affected with moderate IDD and multiple co-occurring behavioral conditions, but their health was generally good or excellent. Many of the individuals resided at home with family members due to a combination of parental and individual desire, as well as the inability to live independently. Our study found that most caregivers of individuals with FXS are generally concerned about multiple aspects of the individual’s future. Caregivers of younger individuals are the most concerned regarding the individual’s future, but also believe they have time before they need to plan or are unable to currently assess the support the individual will need in the future. Caregiver concern does seem to decrease as the individual with FXS ages, which may be due to a better understanding of the diagnosis or the increased functional skills and independence level of older individuals. The results of this study suggest that better understanding of an individual’s FXS diagnosis and greater achieved independence by the individual can relieve some of the concern experienced by caregivers. 

## Figures and Tables

**Figure 1 genes-13-01654-f001:**
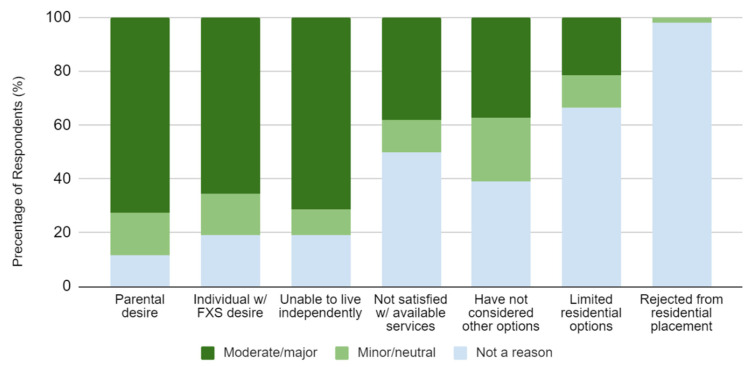
Percentage of caregivers who felt that each factor attributed to the individual with FXS residing at home. There were a total of 57 respondents for “Unable to live independently”. 56 respondents answered for “Parental desire”, “Individual w/FXS desire”, and “Rejected from residential placement”. 55 respondents answered for “Unable to live independently” and “Have not considered other options”. There were 53 total responses for “Not satisfied with available services”.

**Figure 2 genes-13-01654-f002:**
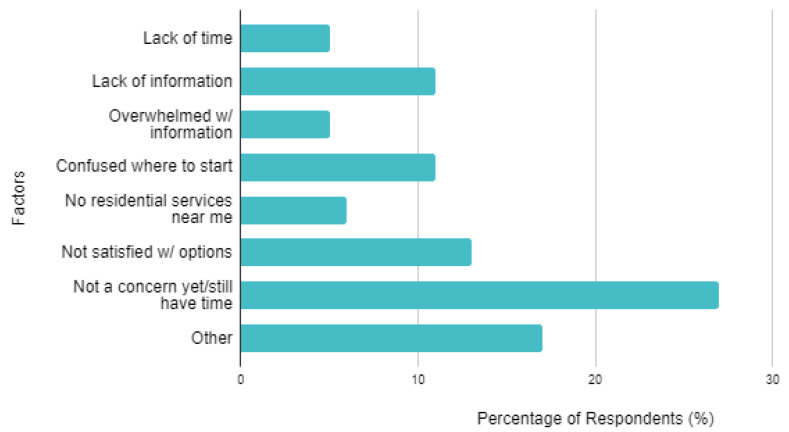
Reasons the caregiver has not started planning or considering alternative living arrangement options, listed by percentage. There were 63 total respondents.

**Figure 3 genes-13-01654-f003:**
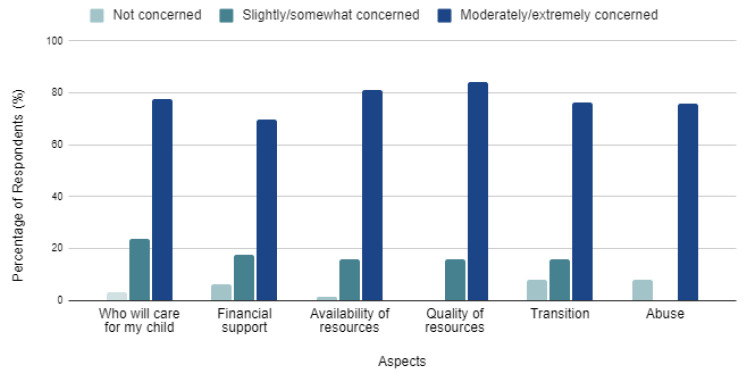
Level of concern regarding aspects of the individual’s future. Total 67 respondents.

**Table 1 genes-13-01654-t001:** Survey Respondent Demographics of Caregivers for Male (*n* = 48) and Female (*n* = 15) Individuals with Fragile X Syndrome.

	Male Female *n* (%)	
Participant relationship to the individual with FXS		
Mother	37 (77%)	14 (93%)
Father	10 (21%)	0 (0%)
Other (sister, caretaker)	1 (2%)	1 (7%)
Caretaker for how many individuals with FXS		
One	42 (88%)	10 (67%)
Two or more	6 (12%)	5 (33%)
Race/Ethnicity		
Caucasian	45 (94%)	14 (93%)
Hispanic/Latino	0 (0%)	0 (0%)
Black	0 (0%)	0 (0%)
Asian	0 (0%)	0 (0%)
American Indian	0 (0%)	0 (0%)
Other	1 (2%)	1 (7%)
Prefer not to answer	2 (4%)	0 (0%)
Region		
Midwest	13 (27%)	6 (40%)
Northeast	10 (21%)	4 (26%)
South	11 (23%)	3 (20%)
West	13 (27%)	1 (7%)
Outside of the U.S.	1 (2%)	1 (7%)
Education, highest level achieved		
Some high school	0 (0%)	1 (7%)
High school degree or equivalent (GED)	4 (9%)	2 (13%)
Technical school, Associate’s degree	11 (23%)	2 (13%)
College degree (Bachelor’s)	16 (33%)	7 (47%)
Post-graduate degree (Master’s/Doctorate)	16 (33%)	3 (20%)
Prefer not to answer	1 (2%)	0 (0%)
Annual household income		
<$50,000	5 (10%)	3 (20%)
$50–100,000	11 (23%)	4 (26%)
$100–150,000	7 (15%)	3 (20%)
$150–200,000	10 (21%)	3 (20%)
>$200,000	7 (15%)	1 (7%)
Prefer not to answer	8 (17%)	1 (7%)
Age group of the individual with FXS		
<15 years	8 (17%)	2 (4%)
15–24 years	20 (42%)	3 (20%)
25–34 years	14 (29%)	6 (40%)
35–44 years	5 (10%)	4 (26%)
45–54 years	1 (2%)	0 (0%)
>55 years	0 (0%)	0 (0%)
Level of Intellectual Developmental Disability		
Mild	2 (4%)	5 (33%)
Moderate	36 (75%)	8 (53%)
Severe	8 (17%)	1 (7%)
No IDD/Unsure	1 (2%)	1 (7%)
No answer	1 (2%)	0 (0%)
Affected with a co-occurring condition:		
Anxiety	46 (96%)	13 (87%)
Mild	12	6
Moderate	21	3
Severe	4	4
Attention problems	47 (98%)	11 (73%)
Mild	12	3
Moderate	28	8
Severe	7	0
Hyperarousal	43 (90%)	4 (27%)
Mild	16	2
Moderate	15	2
Severe	0	0
No answer	0	0
Hyperactivity	36 (75%)	5 (33%)
Mild	18	4
Moderate	14	1
Severe	4	0
Autism Spectrum	32 (67%)	6 (40%)
Mild	13	2
Moderate	15	2
Severe	4	2
No answer	1	0
Intermittent explosive disorder	21 (44%)	3 (20%)
Constant aggressive behavior	19 (40%)	3 (20%)
Self-injury	14 (29%)	1 (7%)
Depression	5 (10%)	7 (47%)
Overall health score		
Excellent	20 (42%)	6 (40%)
Good	27 (56%)	9 (60%)
Fair	1 (2%)	0 (0%)
Poor	0 (0%)	0 (0%)
Very Poor	0 (0%)	0 (0%)

## Data Availability

Raw data were generated at Icahn School of Medicine at Mount Sinai. Derived data supporting finings of this study are available from the corresponding author R.L. on request.

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
