# Peer review of "Exploring Parents’ Concerns Regarding Long-Term Support and Living Arrangements for Their Children with Fragile X Syndrome"

_genes, 2022, doi:10.3390/genes13091654_

Round 1

Reviewer 1 Report

Shuleski et al. presented an important study on the rarely discussed aspect of Fragile X Syndrome. The study is well designed and the results are clearly presented. A drawback is a small cohort.

It would be interesting to know if there are any programs run either by a private institutions or the government to support the independency of the FXS patients?

Also, I wonder if there are any other cognitive scales used for FXS patients?

The article is well written and the results are clearly presented. 

Author Response

Shuleski et al. presented an important study on the rarely discussed aspect of Fragile X Syndrome. The study is well designed and the results are clearly presented. A drawback is a small cohort.

We have added in the discussion: "We would also hope to survey a larger and more ethnically diverse population as cultural factors and social determinants of health likely play a large role in the FXS community."

It would be interesting to know if there are any programs run either by a private institutions or the government to support the independency of the FXS patients?

We are not familiar with public institutions providing independency support specifically for FXS patients. Unfortunately services are limited to government assistance programs for children and adults with intellectual and developmental disabilities as discussed in the text.

Also, I wonder if there are any other cognitive scales used for FXS patients?

Fragile X Syndrome patients, especially adolescents and adults, have been most widely assessed with intellectual quotient (IQ) score. There are some behavioral scales used to assess children with FXS. (Huddleston LB, Visootsak J, Sherman SL. Cognitive aspects of Fragile X syndrome. Wiley Interdiscip Rev Cogn Sci. 2014;5(4):501-508. doi:10.1002/wcs.1296). 

The article is well written and the results are clearly presented. 

Thank you for your review!

Reviewer 2 Report

Dear authors. 

The current manuscript "Exploring parents’ concerns regarding long-term support and living arrangements for their children with Fragile X Syndrome" focuses on an interesting and important topic, the paper is well and clearly written, results are displayed and discussed sufficiently. I have only a minor comment: 

- dependence average score - how exactly was it obtained and calculated?

- why have you not used Pearson correlation analysis for detecting the effect of age on dependence average score (and instead you divided the individuals into several age groups + susequent ANOVA).

- shortcuts IDD and ID are used and IDD is not explained in the text.

Otherwise, I have no other comments and I recommend the manuscript for acceptance.   

Author Response

1) Dependence average score - how exactly was it obtained and calculated

- Survey question D1 was presented as a Likert-scale. These were converted to numerical values and the mean was used as the Dependence Average score. This is outlined in the methods and we have added it to Appendix A for clarity. 

why have you not used Pearson correlation analysis for detecting the effect of age on dependence average score (and instead you divided the individuals into several age groups + susequent ANOVA).

Our questionnaire had respondents select an appropriate age group, rather than raw age. Therefore the data was categorical, and a Pearson correlation could not be used. 

shortcuts IDD and ID are used and IDD is not explained in the text.

We have clarified IDD as Intellectual and Developmental Disabilities in the abstract as well as in the introduction.

Otherwise, I have no other comments and I recommend the manuscript for acceptance.   

Thank you for your review!